# Cascade cyclization of alkene-tethered acylsilanes and allylic sulfones enabled by unproductive energy transfer photocatalysis

Yunxiao Zhang[1,2,6], Yizhi Zhang[1,6], Chen Ye [3,4], Xiaotian Qi [5] ✉, Li-Zhu Wu [3,4] ✉ & Xiao Shen [1,2] ✉

Developing photo-induced cascade cyclization of alkene-tethered acylsilanes is challenging, because acylsilanes are unstable under light irradiation. Herein, we report that the energy transfer from excited acylsilanes to a photocatalyst that possesses lower triplet energy can inhibit the undesired decomposition of acylsilanes. With neutral Eosin Y as the photocatalyst, an efficient synthesis of cyclopentanol derivatives is achieved with alkene-tethered acylsilanes and allylic sulfones. The reaction shows broad substrate scope and the synthetic potential of this transformation is highlighted by the construction of cyclopentanol derivatives which contain fused-ring or bridged-ring.

Organosilicon compounds are important molecules in chemistry and material science, due to their less toxic and easy-to-handle properties and the abundance of silicon element[1-4]. As a kind of unique carbonyl compounds, acylsilanes have been widely used in synthetic chemistry[5-9]. Compared to common ketones, acylsilanes can absorb relatively longer wavelength light, and they usually possess lower triplet energies than ketones (e.g. 3-phenyl-1-(trimethylsilyl)propan-1-one, $E_T = 55.3$ kcal/mol; benzophenone, $E_T = 69.1$ kcal/mol)[10,11]. Although the early studies pioneered by Brook and co-workers focused on the photochemical studies of acylsilanes with UV light[12-14], recent work found that they are labile under visible-light irradiation, sometimes in the presence of a photocatalyst, resulting in the formation of carbenes **A** or acyl radicals **B** (Fig. 1a)[10,15-25]. For example, Priebbenow disclosed a visible-light-induced intramolecular [2 + 1] cycloaddition of acylsilanes with tethered alkenes (Fig. 1b)[23]. Recently, we developed the photocatalyzed intramolecular [2 + 1] cycloaddition of acylsilanes with olefins (Fig. 1c)[24]. In this context, it is challenging to develop a photo-induced cascade cyclization reaction of alkene-tethered acylsilanes with another reaction partner[26-28], because of the undesired decomposition of acylsilanes under light irradiation (Fig. 1a). Glorius[10], Kusama[18] and our group[24,25] have

reported that carbene generation from acylsilanes could be facilitated through the energy transfer catalysis with photocatalysts which possess higher triplet energies than those of the corresponding acylsilanes[29-44]. Inspired by these works, we envisioned that the inhibition of light-induced decomposition of acylsilanes might be achieved via energy transfer from the excited acylsilanes to a photocatalyst that possesses lower triplet energy (Fig. 1d). Under the guidance of this concept, we have developed an efficient photocatalyzed synthesis of cyclopentanol derivatives with alkene-tethered acylsilanes and allylic sulfones (Fig. 1d).

It is worth noting that cyclopentanol is an important structural motif in bioactive molecules. The commercially available drugs such as Levonorgestrel, Faslodex and Remodulin all contain cyclopentanol motif and the sales of these three drugs all exceeded 500 million dollars in 2020[45]. Therefore, the development of novel synthetic methodologies for the construction of cyclopentanol motif is of great value. In recent years, photocatalyzed radical cyclization has emerged as powerful strategy to construct cyclic compounds, and toxic reagents are often not necessary under these conditions[46-51]. However, previous synthesis of cyclopentanols through intramolecular radical cyclization to acylsilanes relied on thermal chemistry[52-54].

[1]The Institute for Advanced Studies, Engineering Research Center of Organosilicon Compounds & Materials, Ministry of Education, Wuhan University, Wuhan, China. [2]Shenzhen Research Institute of Wuhan University, Wuhan University, Shenzhen, China. [3]Key Laboratory of Photochemical Conversion and Optoelectronic Materials, Technical Institute of Physics and Chemistry, Chinese Academy of Sciences, Beijing, China. [4]School of Future Technology, University of Chinese Academy of Sciences, Beijing, China. [5]Engineering Research Center of Organosilicon Compounds & Materials, Ministry of Education, College of Chemistry and Molecular Sciences, Wuhan University, Wuhan, China. [6]These authors contributed equally: Yunxiao Zhang, Yizhi Zhang. ✉e-mail: qi7xiaotian@whu.edu.cn; lzwu@mail.ipc.ac.cn; xiaoshen@whu.edu.cn

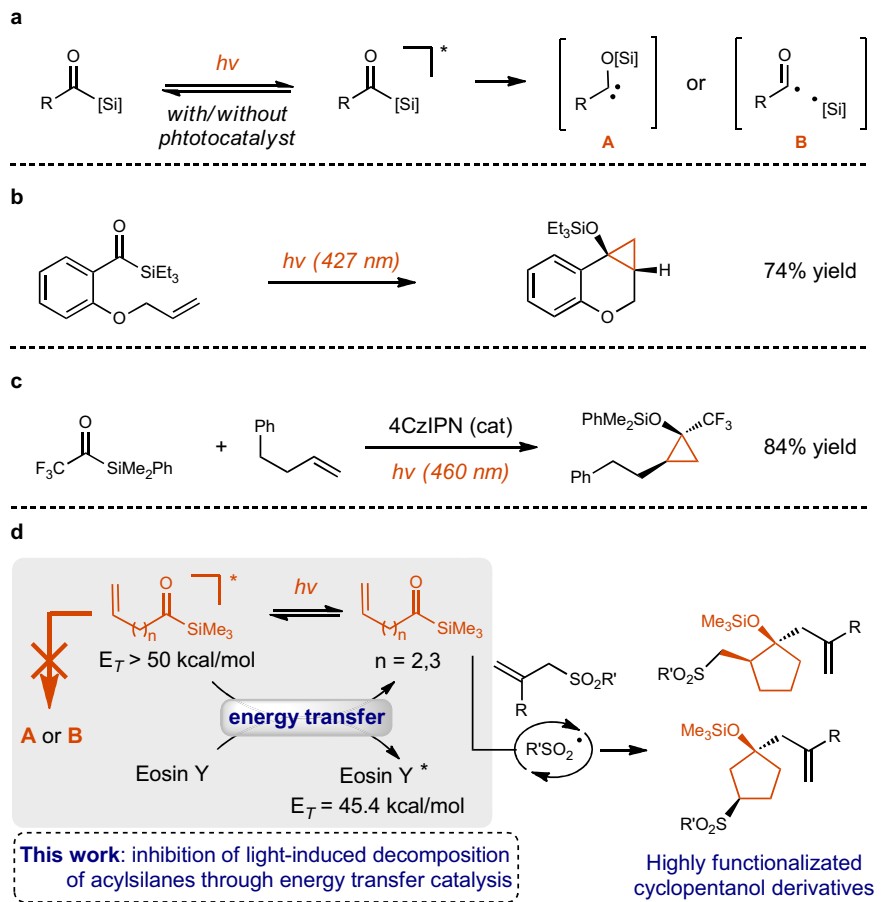

**Fig. 1 | Background and our strategy for the synthesis of cyclopentanols enabled by inhibition of light-induced decomposition of acylsilanes through energy transfer catalysis. a** Previous work: acylsilanes are unstable under light irradiation. **b** Priebbenow's intramolecular [2 + 1] cyclization of acylsilanes with tethered olefins. **c** Our intermolecular [2 + 1] cyclization of acylsilanes with olefins. **d** This work: visible-light-induced cascade cyclization of acylsilanes and allylic sulfones.

In this work, we successfully employed acylsilanes in photocatalyzed multicomponent radical cyclization reaction to synthesize complex cyclopentanols. The reaction shows broad substrate scope and the synthetic potential of this transformation is highlighted by the synthesis of fused-ring and bridged-ring compounds. The success of the reaction is attributed to the energy transfer from excited acylsilanes to the photocatalyst that possesses lower triplet energy which inhibits the undesired decomposition of acylsilanes. This work will pave the way to develop new ground state reactions of acyl silanes under photochemical conditions.

## Results and discussion
### Evaluation of reaction conditions
We commenced our study by testing our proposal about the inhibition of light-induced decomposition of acylsilanes via energy transfer catalysis. Firstly, we tested the conversion of acylsilane **1a** in CDCl₃ (Fig. 2a). It was found that **1a** gradually decomposed under white light. However, the addition of 1 mol% of neutral Eosin Y dramatically decreased the conversion of **1a**. The control experiment confirmed that **1a** is stable in dark conditions and no conversion was observed in the absence of light after 30 h. The results of luminescence quenching experiments confirmed that the excited state of **1a** could be quenched with different concentrations of Eosin Y (Fig. 2b). We can see from the UV-vis spectrum that there is no significant overlap between **1a** and Eosin Y absorbance (Fig. 2c). Moreover, transient absorption measurements have been conducted to investigate the possible triplet-triplet energy transfer from **1a** to Eosin Y. We found that the sample

which contain both **1a** and Eosin Y showed strong $T_1 \rightarrow T_n$ transition of Eosin Y at around 580 nm, which was much stronger than the signal in the sample with only Eosin Y. These results strongly support the involvement of triplet-triplet energy transfer process between **1a** and Eosin Y, which is consistent to the lower triplet energy of neutral Eosin Y ($E_T = 43.6$ kcal/mol)[55] than that of acylsilane **1a** ($E_T = 56.7$ kcal/mol) (For the details on the calculation of the triplet energy of acylsilane **1a**, see the Supporting Information.). Encouraged by the above results, we then applied the concept of inhibition of light-induced decomposition of acylsilanes via energy transfer from excited acylsilanes to neutral Eosin Y in the synthesis of cyclopentanol derivatives. The investigation of reaction conditions was performed with alkene-tethered acylsilane **1a** and allylic sulfone **2a** as the model substrates under photocatalysis conditions (Table 1). We found that when neutral Eosin Y (1 mol%) was used as the catalyst in the presence of KOPiv (1 equiv.) as base, the reaction between acylsilane **1a** (2 equiv.), allylic sulfone **2a** (1 equiv.) and PhSO₂Na (0.2 equiv.) in MeCN/H₂O (v/v = 1/3, 0.067 M) at room temperature under white LEDs (6 W) for 12 h generated product **3a** in 89% yield with 90/10 dr (isolated in 82% yield, Table 1, entry 1). Cyclohexanol derivative **4a**, acylation product **5a** and [2 + 1] cyclization product **6a** were not detected. Without neutral Eosin Y, there was only 14% yield of **3a**, while 91% conversion of **1a** was observed, indicating the importance of the neutral Eosin Y in the protection of **1a** from undesired decomposition (Table 1, entry 2). The use of Ru(bpy)₂Cl₂ ($E_T = 46.5$ kcal/mol, $E_{1/2}$(*Ru^II/Ru^I) = 0.77 V in MeCN vs SCE)[29] as the photocatalyst resulted in decreased yield of **3a** and conversion of **1a**, probably because of its low oxidation potential and the lower triplet energy

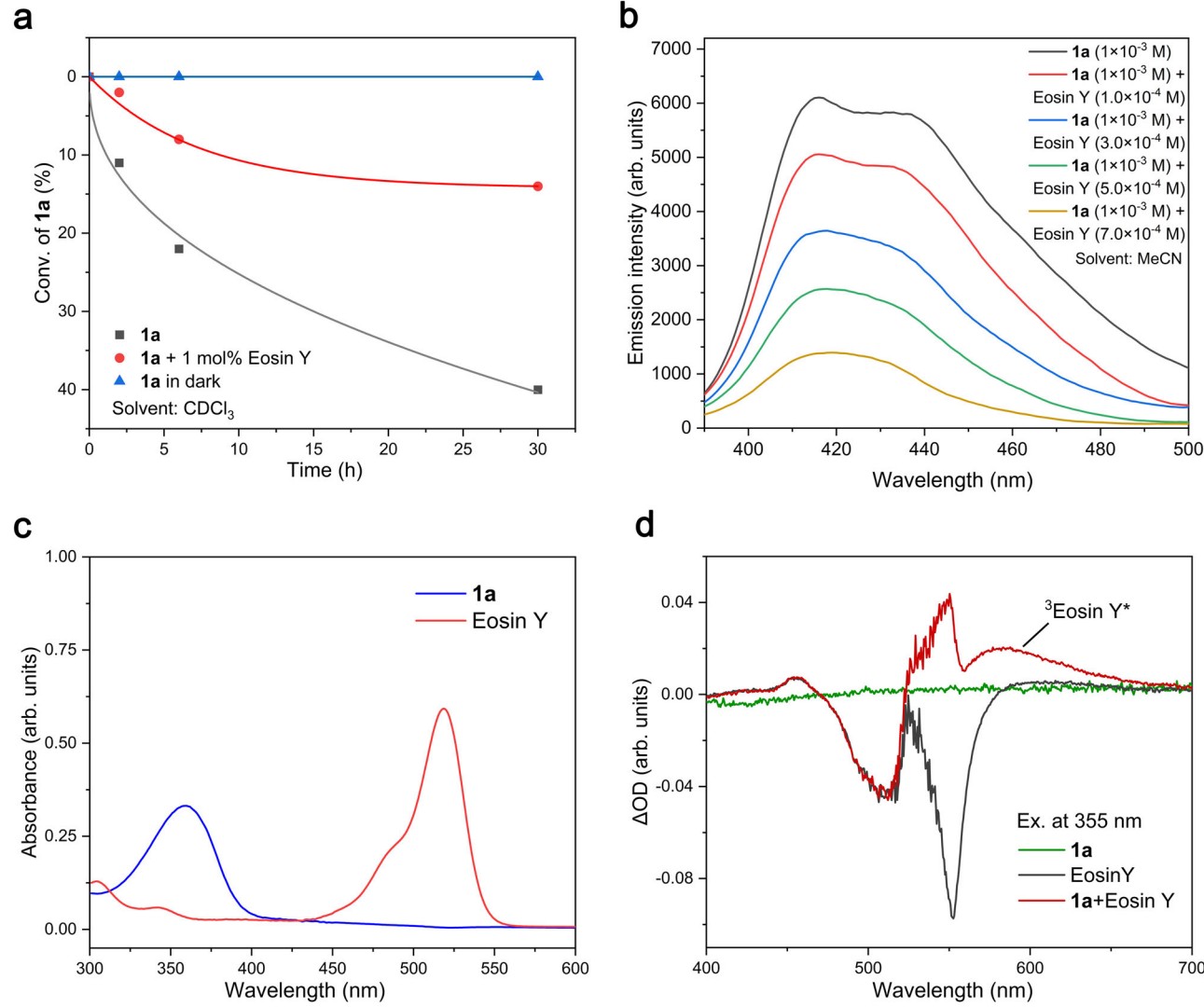

**Fig. 2 | Proof of concept: inhibition of visible-light-induced decomposition of acylsilanes via energy transfer catalysis. a** Light-induced decomposition of **1a**. **b** Luminescence quenching experiments. **c** UV/Vis of **1a** and Eosin Y. **d** Transient absorption spectra observed after laser excitation ($\lambda_{ex}$ = 355 nm) of system containing (green line) 1 mM **1a** (black line), 0.1 mM Eosin Y and (red line) 1 mM **1a** + 0.1 mM Eosin Y. Source data are provided as a Source Data file.

than acylsilane **1a**, (Table 1, entry 3). The use of *fac*-Ir(ppy)$_3$ ($E_T$ = 57.8 kcal/mol)[29], 4CzIPN ($E_T$ = 59.6 kcal/mol)[56] or FIrPic ($E_T$ = 60.5 kcal/mol)[26] as the photocatalyst resulted in decreased yield of **3a**, probably because their relatively higher triplet energies than that of **1a** would promote the decomposition of **1a** to carbene intermediate (Table 1, entries 4~6). Meanwhile, the formation of acylation product **5a** could be explained by the oxidation-induced C-Si bond homolysis to generate acyl radical, due to the higher oxidation ability of excited 4CzIPN and FIrPic (4CzIPN, $E_{1/2}$(PC*/PC$^{\bullet-}$) = 1.43 V in MeCN vs SCE;[57] FIrPic, $E_{1/2}$(*Ir$^{III}$/Ir$^{II}$) = 1.40 V in MeCN vs SCE[26]; neutral Eosin Y, $E_{1/2}$($^3$EY*/EY$^-$) = 0.83 V in MeCN vs SCE[58]; **1a**, $E_{1/2}$ = 1.23 V in MeCN vs SCE [For the details on the cyclic voltammograms of the acylsilanes used in this study, please see the Supporting Information]). The above proposal was further supported by the experimental result with 9-mesityl-10-methylacridinium perchlorate ($E_T$ = 44.7 kcal/mol[59]; $E_{1/2}$(PC*/PC$^{\bullet-}$) = 2.06 V in MeCN vs SCE[60]) as the catalyst, in which, significantly decreased conversion of **1a** and increased yield of **5a** were obtained (Table 1, entry 7). Control experiments revealed that the use of MeCN/H$_2$O mixture was important for the high yield of the reaction (Table 1, entries 8 and 9). Without light, there was less than 2% conversion of acylsilane **1a** (Table 1, entry 10). PhSO$_2$Na and KOPiv were not necessary for the success of the reaction, but the addition of them indeed

increased the efficiency, which might because they can promote the generation of PhSO$_2$ radical (Table 1, entries 11 and 12). Further study confirmed that the reaction is sensitive to air, supporting that the reaction might proceed through a radical mechanism (Table 1, entry 13).

## Scope of the reaction

With the optimal reaction conditions in hand, we investigated the reaction scope for the synthesis of $\beta$-substituted cyclopentanol derivatives with acylsilane **1a** as the reagent (Fig. 3). Firstly, we tested the influence of silyl groups on the efficiency of the reaction. The small SiMe$_3$ was found to be better than SiEt$_3$, SiMe$_2$(*t*-Bu), SiMe$_2$Ph and SiMePh$_2$, although all of them afforded the desired product in more than 50% yield, and the diastereoselectivity was not significantly affected. A variety of allylic sulfones could be employed as the substrates, affording compounds **3f-3aa** in 52%-95% yield, with up to 99/1 dr. The Csp$^2$-Me, Csp$^2$-F, Csp$^2$-Cl, Csp$^2$-Br, Csp$^2$-I, Csp$^2$-OMe, Csp$^2$-CF$_3$, Csp$^3$-CN bonds were tolerated, affording products **3f-3o** and **3t** in 55-95% yield. In addition, cyclopropane-containing compound **3p** was synthesized in 88% yield with 99/1dr. Electron-rich thienyl and furyl groups were tolerated (**3q**, 82% yield, 90/10 dr; **3r**, 85% yield, 89/11 dr). Moreover, primary alcohol (**3w**, 82% yield, 90/10 dr), tertiary alcohol

**Table 1 | Investigation of reaction conditions[a]**

| Entry | Variation from the "standard" conditions | Conversion of 1a [%] | Yield of 3a [%] (dr) | Yield of 5a [%] (dr) |
|---|---|---|---|---|
| 1 | None | 88 | 89 (90/10), 82[b] | 0 |
| 2 | No neutral Eosin Y | 91 | 14 (86/14) | 0 |
| 3 | Ru(bpy)₂Cl₂, instead of neutral Eosin Y | 15 | 20 (90/10) | 0 |
| 4 | fac-Ir(ppy)₃, instead of neutral Eosin Y | 84 | 38 (80/20) | 0 |
| 5 | 4CzIPN, instead of neutral Eosin Y | 80 | 40 (90/10) | 4 |
| 6 | FIrPic, instead of neutral Eosin Y | 90 | 8 (87/13) | 2 |
| 7 | [Acr⁺-Mes]ClO₄⁻, instead of neutral Eosin Y | 18 | 12 (84/16) | 16 |
| 8 | MeCN, instead of MeCN/H₂O = 1/3 (v/v) | 77 | 46 (87/13) | 0 |
| 9 | H₂O, instead of MeCN/H₂O = 1/3 (v/v) | 80 | 4 (87/13) | 0 |
| 10 | No light | <2% | 0 | 0 |
| 11 | No PhSO₂Na | 47 | 56 (90//10) | 0 |
| 12 | No KOPiv | 87 | 78 (90/10) | 0 |
| 13 | Air atmosphere | 88 | 10 (86/14) | 0 |

[a]Reactions were run on 0.1 mmol scale in 1.5 mL of solvent for 12 h under N₂. The conversion, yield and dr were determined by ¹H NMR spectroscopy with BrCH₂CH₂Br as an internal standard.
[b]Yield in parentheses refers to isolated yield of two diastereoisomers, and the relative configuration of the major isomer is shown.

(**3x**, 88% yield, 92/8 dr), aliphatic aldehyde (**3 y**, 80% yield, 90/10 dr) have been successfully synthesized, further highlighting the functional group tolerance of the reaction. Two examples of the synthesis of bioactive molecular derivatives have also been achieved, indicating the synthetic potential of the reaction (**3z**, 56% yield, 80/20 dr, **3aa**, 52% yield, 93/7 dr).

It is worthy to note that the cyclopentyl siloxanes could be easily converted to cyclopentanols through the desilylation with TBAF, and **3a'** was obtained in 97% yield with 90/10 dr with **3a** as the starting material. The relative configuration of the major isomer of **3j** was determined by X-Ray Crystal Structure Analysis[61], and the configuration of other products were assigned accordingly.

After achieving the efficient synthesis of β-substituted cyclopentanol derivatives with **1a** as the reagent, we applied the cascade cyclization reaction in the synthesis of γ-substituted cyclopentyl siloxanes with acylsilane **1 f** (Fig. 4). Various five-membered ring products were synthesized in moderate yields, although the diastereoselectivities were generally low (**7a-7i**, 48%-71% yield, 50/50-65/35 dr). Again, the reaction tolerated Csp²-Me, Csp²-F, Csp²-Cl, Csp²-Br, Csp²-I, Csp²-OMe, Csp²-CF₃ bonds. Electron-rich furyl group-containing compound **7 g** was isolated in 71% yield, and the naphthyl-group containing product **7 h** was obtained in 51% yield. In all cases, we did not observe any four-membered ring product. When TBAF was used to quench the reaction, cyclopentanol **7a'** was isolated in 60% yield, 53/47 dr.

Since only catalytic amount of PhSO₂Na was needed to facilitate the above reaction (Fig. 3), we wondered whether our reaction could be applied in the synthesis of different functionalized sulfones by modification of the allylic sulfones. We found that the reactions performed efficiently for the synthesis of β-substituted cyclopentyl siloxanes in the presence of only 2-5 mol% of PhSO₂Na (Fig. 5). Compounds **3ab**-**3ag** were synthesized in 54-87% yield with 84/16-95/5 dr. However, 2 equivalents of ArSO₂Na was needed to promote the

reaction for the synthesis of γ-substituted cyclopentyl siloxanes, indicating their less efficiency in the completion of the catalytic cycles.

## Synthetic transformations of compound 3a and 7a

It is worthy to note that cyclopentanols containing fused-ring and bridged-ring motifs are widely found in bioactive molecules. For example, compound Y is a Na⁺K⁺-ATPase inhibitor and compound Z is an adenosine receptor antagonist[62,63]. Taking the advantage of the strong electron-withdrawing group ability of the sulfonyl group, the deprotonation-conjugate addition reaction of compound **3a** has been achieved, affording fused-ring compound **8** in 82% yield with >99:1 dr. With compound **7a** as the substrate, bridged-ring compound **9** was synthesized in 74% yield with >99:1 dr (Fig. 6). The facile construction of these two types of compounds with the cyclization products demonstrated the synthetic potential of the current methodology.

## Mechanistic study

Preliminary studies have been conducted to understand the possible mechanism (Fig. 7). The addition of 1 equivalent of TEMPO completely inhibited the reaction of **1a** and **2a**, and compound **X** was detected by HRMS, indicating that PhSO₂ radical might participated in the catalytic cycle (Fig. 7a). The time profile of the light on/off experiments over time supported that the reaction might proceed through radical chain reaction (Fig. 7b). In addition, the reaction could proceed in dark after light irradiation for 10 min (Fig. 7b). The quantum yield around 1.5 also supported the radical chain mechanism (Fig. 7b). The Stern-Volmer fluorescence-quenching experiments validated the interaction of neutral Eosin Y with both allylic sulfone **2a** and PhSO₂Na, which were two possible pathways to generate PhSO₂ radical (Fig. 7c). However, we cannot rule out the possibility of generation of PhSO₂ radical through the sensitization of **2a** with excited acylsilane or direct homolysis of **2a** under light irradiation. In addition, PhSO₂Na can be oxidized by

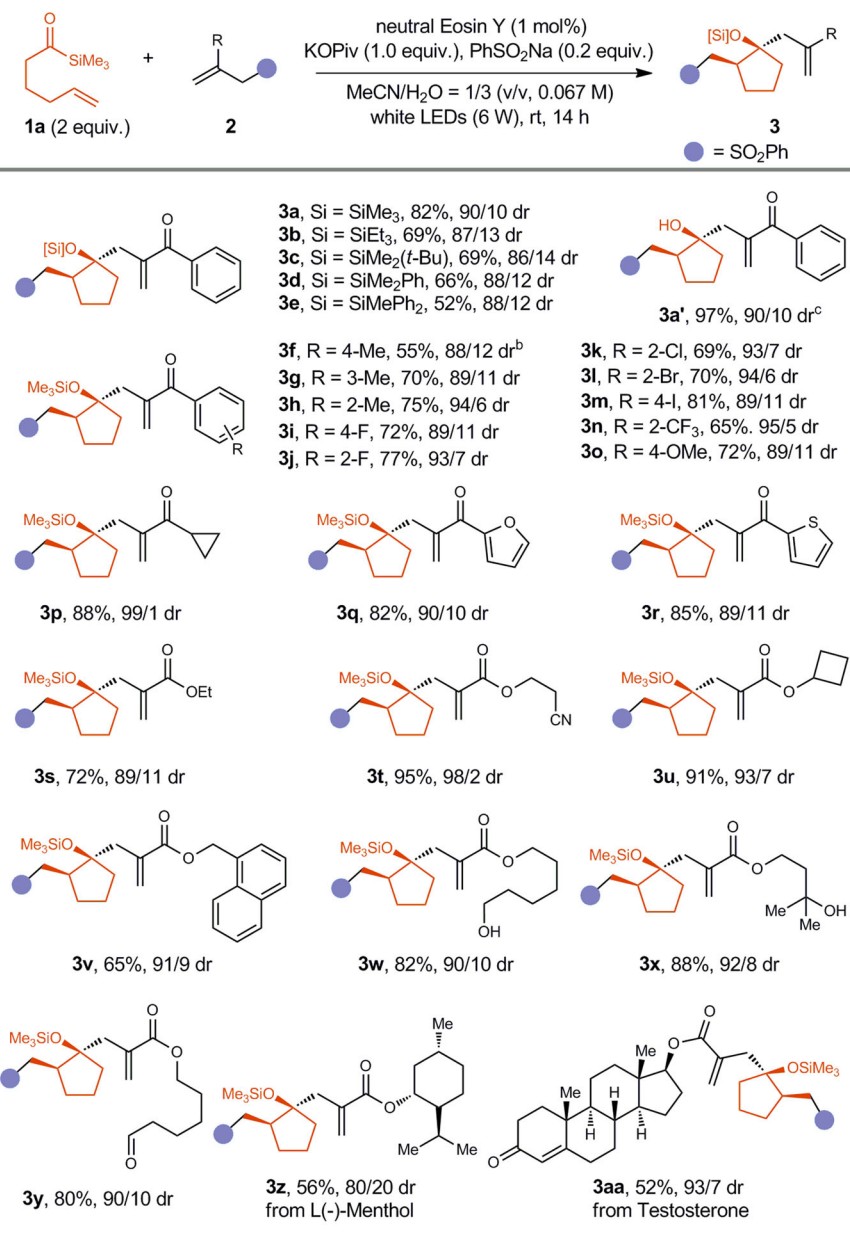

**Fig. 3 | Scope for the synthesis of β-substituted cyclopentanol derivatives. a** Total yield of the isolated two diastereoisomers is reported. **b** neutral Eosin Y (2 mol%), KOPiv (1.5 equiv.), 36 h. **c** 3a (0.05 mmol), TBAF (1 M in THF, 0.1 mmol), rt, 0.5 h.

excited neutral Eosin Y to generate PhSO$_2$ radical[64]. The importance of ArSO$_2$Na in the reaction for the synthesis of γ-substituted cyclopentanol derivatives indicated that ArSO$_2$ radical was more likely generated from ArSO$_2$ anion first. When *E*-stilbene ($E_T$ = 51.0 kcal/mol)[65] was added to the reaction, only 24% yield of cascade cyclization product was obtained and the isomerization of *E*-stilbene to *Z*-stilbene was observed, supporting the energy transfer from excited acylsilane to *E*-stilbene (Fig. 7d).

Density functional theory (DFT) calculations were then carried out to study the radical chain mechanism and the selectivity control in the cascade cyclization of alkene-tethered acylsilane **1a** with allylic sulfone **2a**. As shown in Fig. 8, the radical addition of PhSO$_2$ radical to the terminal carbon of alkene (via **TS-1a**) generates a secondary carbon-centered radical **10a**. The activation free energy is 3.2 kcal/mol lower than the formation of primary carbon-centered radical **10b** (via **TS-1b**). Radical cyclization (via **TS-2a** or **TS-2b**) is demonstrated to be the

regioselectivity-determining step. The irreversible formation of five-membered ring through **TS-2a** is kinetically favored due to the higher stability of the secondary carbon-centered radical. From oxygen-centered radical **11a**, the radical Brook rearrangement occurs easily through **TS-3a** ($\Delta G^\ddagger$ = 3.8 kcal/mol), leading to the formation of a stable tertiary carbon-centered radical. Subsequent Giese-type radical addition towards electron-deficient alkene **2a** controls the diastereoselectivity. The steric clash evidenced by the H•••H distance of 2.23 Å in **TS-4b** renders the formation of radical **13b** kinetically disfavored ($\Delta\Delta G^\ddagger$ = 4.7 kcal/mol), resulting in the selective formation of *trans* product. Finally, the β-scission of alkyl radical **13a** generates the five-membered cyclization product **3a** and regenerates the PhSO$_2$ radical.

Based on these experimental and computational results, a possible mechanism was proposed in Fig. 9. The energy transfer from the excited acylsilane to neutral Eosin Y inhibited the undesired decomposition of acylsilane **1a** under the photocatalysis conditions and keep

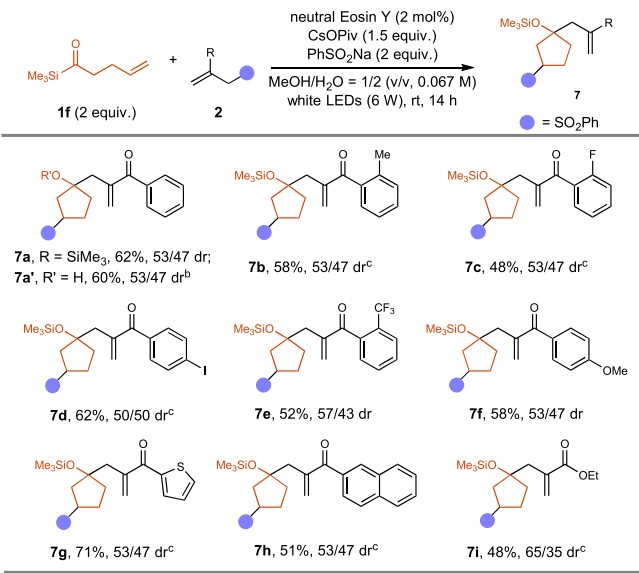

**Fig. 4 | Scope for the synthesis of γ-substituted cyclopentanol derivatives. a** Total yield of the isolated two diastereoisomers is reported. **b** Yield of the alcohol after desilylation with TBAF is reported. **c** neutral Eosin Y (2 mol%), KOPiv (1.5 equiv.), 36 h.

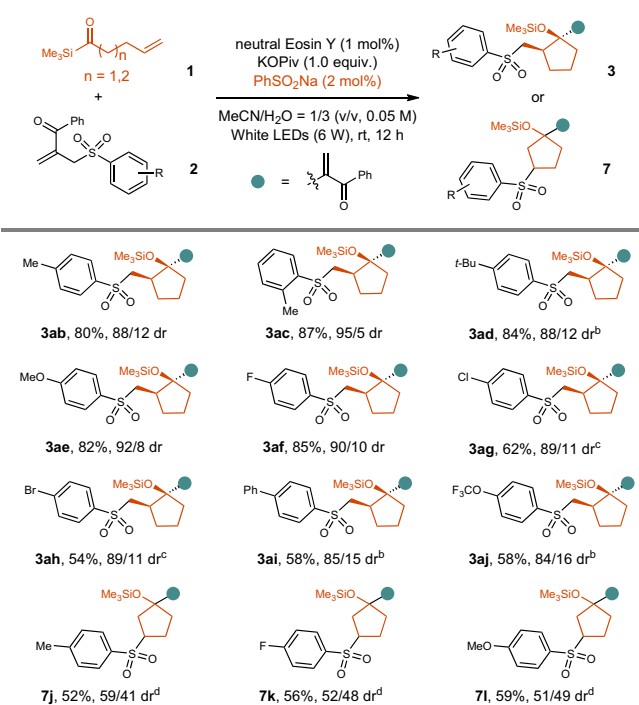

**Fig. 5 | Scope for the synthesis of different sulfonyl group-substituted cyclopentanol derivatives. a** Total yield of the isolated two diastereoisomers is reported. **b** ArSO₂Na (20 mol%) was used. **c** PhSO₂Na (5 mol%) was used. **d** ArSO₂Na (2 equiv.) was used, MeOH/H₂O = 1/2 (v/v).

enough acylsilane in ground state to participate in the reaction with PhSO$_2$ radical. The generation of more stable secondary carbon radical **10a** is favorable over primary carbon radical **10b**. The intramolecular addition of **10a** to the acylsilane would generate alkoxyl radical **11**, followed by radical Brook rearrangement to generate **12**[66–76]. Giese-type addition of **12** to allylic sulfone would generate intermediate **13**, which would eliminate PhSO$_2$ radical and product **3a** to close mechanism cycle. Mechanistic studies revealed that the formation of five-membered cyclization product is promoted by the higher stability of secondary carbon-centered radical and the formation of *trans* product is promoted by the steric repulsion in Giese addition. As for

the reactions with **1 f**, 2 equivalents of ArSO$_2$Na was needed to get high yield, indicating that radical chain process was less efficient than the reaction with **1a** (for the DFT calculations and the proposed mechanism, see supporting information).

In summary, we have developed a visible-light-induced cascade cyclization reaction of alkene-tethered acylsilanes with allylic sulfones. The reaction shows broad substrate scope, enabling the synthesis of various β-substituted and γ-substituted cyclopentanol derivatives. The synthetic potential of the transformation is highlighted by the construction of fuse-ring and bridge-ring compounds. The success of the reaction is attributed to the inhibition of undesired decomposition of

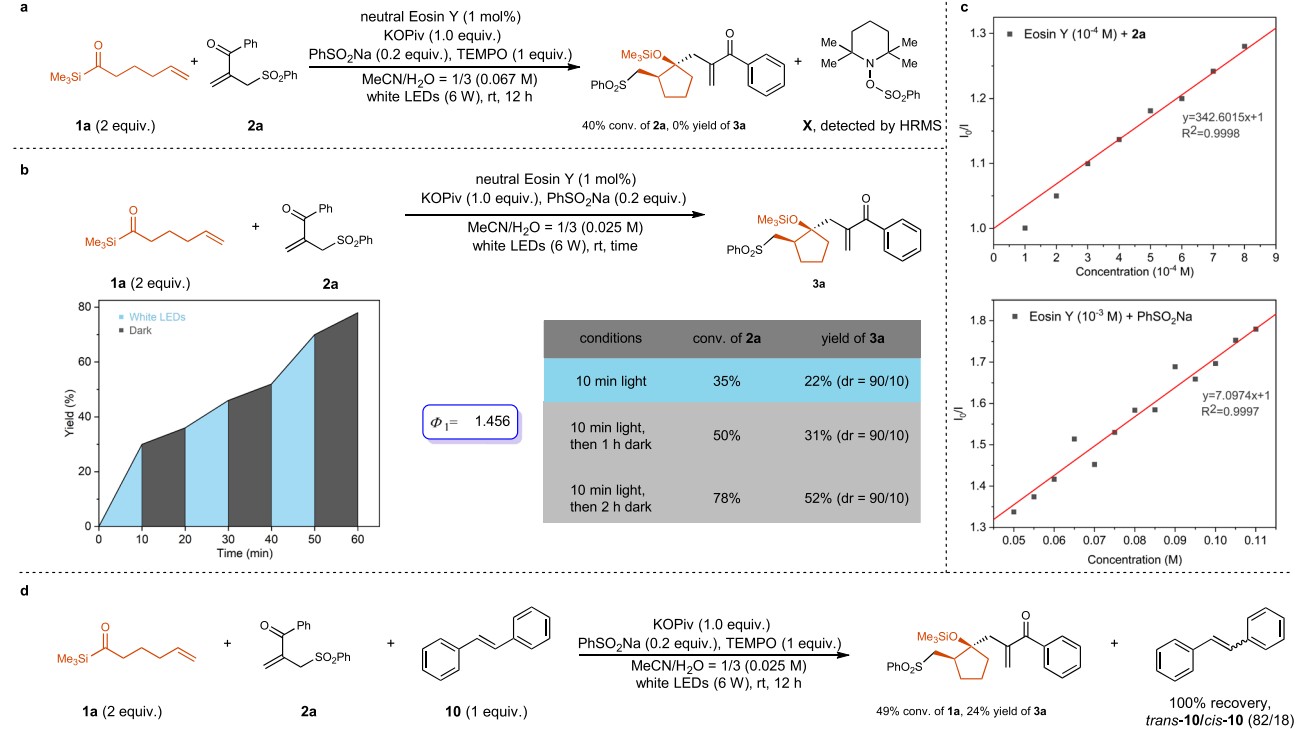

**Fig. 6 | Down-stream transformations.** Fused ring compound **8** and bridged ring compound **9** were successfully prepared with cyclization products **3a** and **7a**, respectively.

**Fig. 7 | Mechanism study. a** Reaction in the presence of TEMPO indicates that PhSO₂ radical might participated in the reaction. **b** Light on-off experiments, determination of quantum yield and light-darkness control experiment support that the reaction might proceed through radical chain mechanism. **c** Eosin Y emission quenching by **2a** and Eosin Y emission quenching by PhSO₂Na. The Stern-Volmer fluorescence-quenching experiments validate the interaction of neutral Eosin Y with both allylic sulfone **2a** and PhSO₂Na. **d** Reaction in the presence of *E*-stilbene instead of Eosin Y supports the energy transfer from excited acylsilane to *E*-stilbene. Source data are provided as a Source Data file.

acylsilanes through energy transfer from excited acylsilanes to the photocatalyst which possesses lower triplet energy. Previously, energy transfer has been mainly used in facilitating excitation state chemistry of acylsilanes, the strategy disclosed here would contribute to the design of more ground state reaction of acylsilanes under photocatalysis conditions.

## Methods
### Typical procedure 1
(**3a**). In a glovebox, to an oven-dried 10 mL tube was added **2a** (28.6 mg, 0.1 mmol), PhSO₂Na (3.2 mg, 0.02 mmol, 0.2 equiv.), Eosin Y (0.65 mg, 0.001 mmol, 1 mol%), KOPiv (14.2 mg, 0.1 mmol,

1 equiv.), MeCN/H₂O = 1:3 (0.067 M) and **1a** (34.0 mg, 0.2 mmol, 2 equiv.) sequentially. The tube was sealed, then irradiated with 6 W white LED lamps. The mixture was stirred under white light irradiation at ambient temperature for the 12 h. Then the light was turned off. The resulting mixture was filtered through a thin silica gel plug with EA (30 mL) as the eluent. The organic phase was concentrated under reduced pressure. The dr was determined by the analysis of the unpurified crude mixture by ¹H NMR. The crude product was purified with column chromatography on silica gel (300~400 mesh) with PE/EA = 5/1 (v/v) as eluent to afford the title compound as a colorless oil (37.5 mg, 82% yield in total, a mixture of two disteraosiomers).

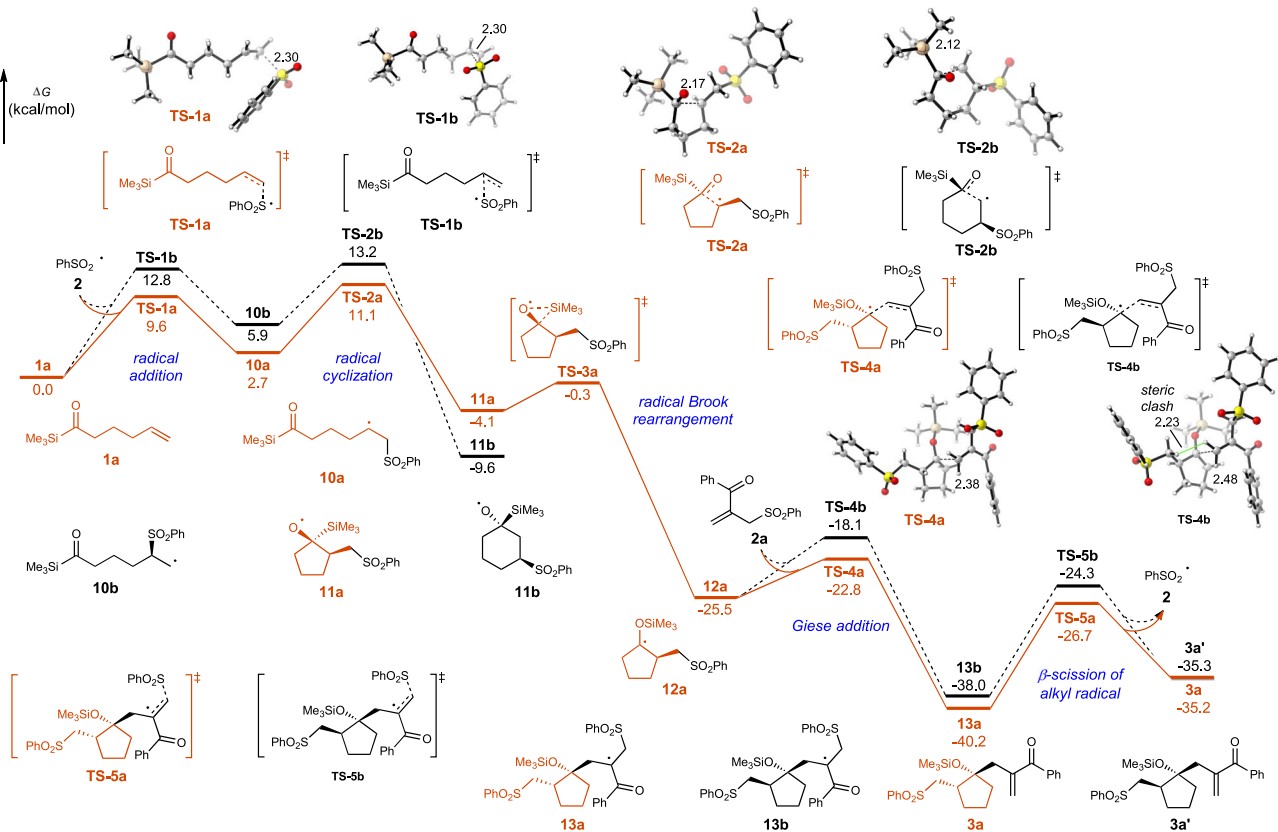

**Fig. 8 | Computational study of the radical chain mechanism in PhSO₂ radical involved cascade cyclization of alkene-tethered acylsilane 1a and allylic sulfone 2a.** All energies were calculated at M06-2X/6-311 + G(d,p)/SMD(water)//M06-2X/6-31 G(d) level of theory.

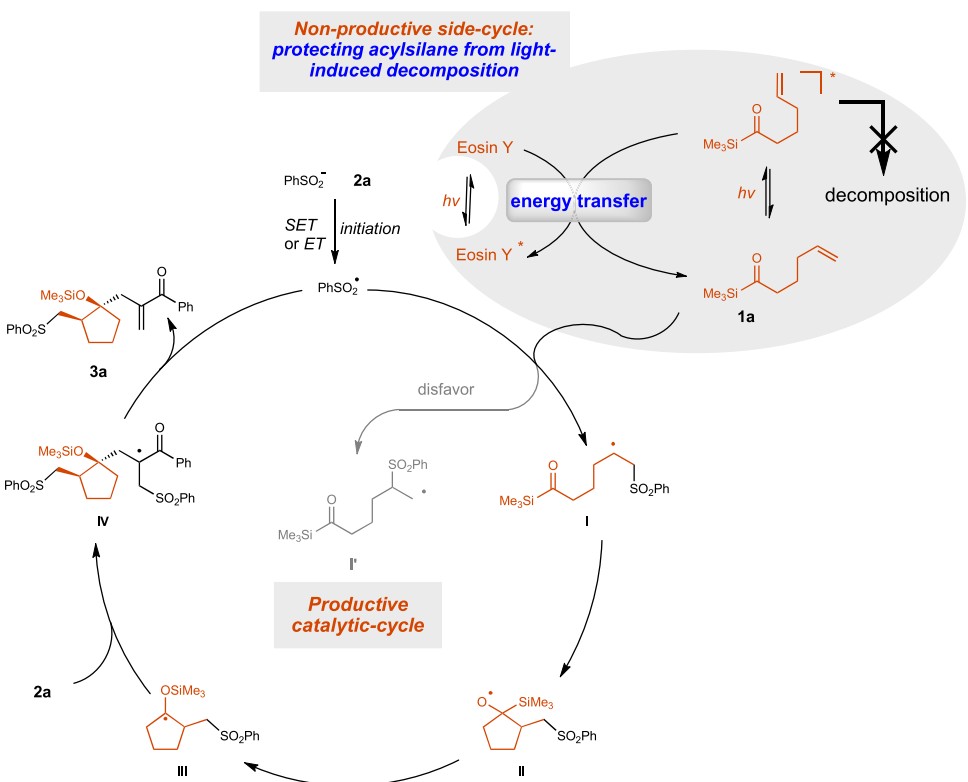

**Fig. 9 | Proposed mechanism.** The unproductive energy transfer from excited acylsilane to Eosin Y inhibited the light-induced decomposition of acylsilanes, enabling the desired cascade cyclization reaction to generate cyclopentanol derivatives.

## Typical procedure 2

**(7a).** In a glovebox, to an oven-dried 10 mL tube was added **2a** (28.6 mg, 0.1 mmol), PhSO$_2$Na (32.8 mg, 0.2 mmol, 2 equiv.), Eosin Y (1.3 mg, 0.002 mmol, 2 mol%), CsOPiv (35.1 mg, 0.15 mmol, 1.5 equiv.), MeOH/H$_2$O = 1:2 (0.067 M) and **1f** (31.2 mg, 0.2 mmol, 2 equiv.) sequentially. The tube was sealed, then irradiated with 6 W white LED lamps. The mixture was stirred under white light irradiation at ambient temperature for the 12 h. Then the light was turned off. The resulting mixture was filtered through a thin silica gel plug with EA (30 mL) as the eluent. The organic phase was concentrated under reduced pressure. The crude product was purified with column chromatography on silica gel (300~400 mesh) with PE/EA = 5/1 (v/v) as eluent to afford the title compound as a colorless oil (27.5 mg, 62 % yield in total, a mixture of two diastereoisomers).

## Data availability

Source data are provided with this paper. The authors declare that all other data supporting the findings of this study are available within the article and Supplementary Information files, and also are available from the corresponding author on request.

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

## Acknowledgements

We are grateful to the National Natural Science Foundation of China (21901191, XS), Guangdong Basic and Applied Basic Research Foundation (2021A1515010105, XS; 2021A1515110248, Yunxiao Z.) and the Fundamental Research Funds for the Central Universities (XS) for financial support. The theoretical calculations were performed on the supercomputing system in the Supercomputing Center of Wuhan University.

## Author contributions

X.S. designed and directed the investigations and composed the manuscript with revisions provided by the other authors. Yunxiao Z. developed the catalytic method. Yunxiao Z. and Yizhi Z. studied the substrate scope. Yizhi Z. performed the DFT calculations. X.Q. directed the calculations. C.Y. performed the transient absorption measurements. L.Z.W. directed the transient absorption measurements. All the authors were involved the analysis of results and discussions of the project.

## Competing interests

The authors declare no competing interests.
