## [Peer Review File · Nature Communications]

Cascade Cyclization of Alkene-tethered Acylsilanes and Allylic Sulfones Enabled by Unproductive Energy Transfer PhotocatalysisREVIEWER COMMENTS

Reviewer #1 (Remarks to the Author):

The authors report the photochemical synthesis of carbocyclic scaffolds from alkene-tethered acyl silanes and allylic sulfones. This work represents a significant advancement in acyl silane chemistry as the photochemical reaction can be realised without activation of the acyl silane which under light irradiation typically generates acyl radical or carbene intermediates.

In this manuscript, the authors describe an innovative approach involving unproductive energy transfer between an acyl silane and photocatalyst to avoid such activation of the acyl silane, leaving sufficient acyl silane intact to react as an electrophile in a related radical cascade process. These findings are expected to pave the way for others to develop new ground state reactions of acyl silanes under photochemical conditions.

There is an appropriate amount of mechanistic and computational data to support the authors claims and I would recommend publication after addressment of the following points:

(1) While the introduction covers previous works regarding acyl silane chemistry, there is no mention of the background of the actual radical chain process that forms the basis of this paper. This manuscript would be strengthened if the authors could discuss the background to the photocatalysed radical cyclisation process. Is this a well-established reaction - what are the challenges and design strategies behind this choice of reaction?

(2) Are the authors able to confirm the emission wavelengths of the white LEDs used? How much of the emission band overlaps with the absorption of the acyl silane - it appears unusual that only 1 mol% of photocatalyst is sufficient to quench the acyl silane absorption which would indicate that there is not a huge amount of emission at from the white LED that corresponds to the absorption of the acyl silane.

Reviewer #2 (Remarks to the Author):

This article by Shen and coworkers describes an efficient photocatalytic synthesis of cyclopentanol derivatives from the reaction of alkene-tethered acylsilanes with allylic sulfones in the presence of Eosin Y under blue irradiation.

The broad scope of the substrate, along with the high reaction efficiency, are excellent features of this method.

The study is well conducted, the SI is complete, and the results clearly presented.

I think this work deserves to be considered for publication in Nature Comm. However, some important mechanistic concerns should first be addressed before acceptance of this work.

1) As stated by the authors, one of the most challenging aspects of their reaction is the inhibition of the photochemical decomposition of acylsilanes by using photosensitizers with low triplet energy. In this context, Eosin Y (ET = 43.6 kcal/mol) was selected as an ideal photocatalyst. However, this problem can also be circumvented by simply irradiating at 530 nm as at this wavelength only Eosin Y will be excited. Looking at the supporting information (Table S2; entry 5), I was surprised to see that the reaction didn't proceed under green light irradiation.

2) The luminescence quenching experiment depicted in Figure 2 tells only that the excited state of the Eosine is quenched with different concentrations of 1a. However, it does not confirm the involvement of triplet energy transfer as claimed by the authors. I believe only transient spectroscopy experiments could confirm the occurrence of such a process.

3) As a minor note, the authors should add a recent review on radical Brook rearrangements recently published in *Angewandte Chemie* (DOI: 10.1002/ange.202205671).

In fine, in this work, Shen and colleagues report a nice photochemical transformation that will certainly be of interest to both synthetic and physical organic chemists. However, they should provide strong mechanistic evidences of the claimed mechanism.

Respond to the reviewer's comments

Reviewers' Comments:

Referee: 1

Reviewer #1: (Remarks to the Author):

The authors report the photochemical synthesis of carbocyclic scaffolds from alkene-tethered acyl silanes and allylic sulfones. This work represents a significant advancement in acyl silane chemistry as the photochemical reaction can be realised without activation of the acyl silane which under light irradiation typically generates acyl radical or carbene intermediates.

In this manuscript, the authors describe an innovative approach involving unproductive energy transfer between an acyl silane and photocatalyst to avoid such activation of the acyl silane, leaving sufficient acyl silane intact to react as an electrophile in a related radical cascade process. These findings are expected to pave the way for others to develop new ground state reactions of acyl silanes under photochemical conditions.

There is an appropriate amount of mechanistic and computational data to support the authors claims and I would recommend publication after addressment of the following points:

Response: Thanks for the positive recommendation.

1) While the introduction covers previous works regarding acyl silane chemistry, there is no mention of the background of the actual radical chain process that forms the basis of this paper. This manuscript would be strengthened if the authors could discuss the background to the photocatalysed radical cyclisation process. Is this a well-established reaction - what are the challenges and design strategies behind this choice of reaction?

Response: Thanks for the kind suggestion. We have discussed the background of photocatalyzed radical cyclisation in the revised manuscript and recent reviews and selected papers on this topic have been cited as refs 46~51. We also included the previous thermal synthesis of cyclopentanols through intramolecular radical cyclization to acylsilanes (refs 52~54). The following sentences have been added in the revised introduction: “In recent years, photocatalyzed radical cyclization has emerged as powerful strategy to construct cyclic compounds, and toxic reagents are often not necessary under these conditions.⁴⁶⁻⁵¹ However, previous synthesis of cyclopentanols through intramolecular radical cyclization to acylsilanes relied on thermal chemistry.⁵²⁻⁵⁴ The successful employment of acylsilanes in photocatalyzed multicomponent radical cyclization to synthesize complex cyclopentanols which is disclosed in this work would pave the way to develop new ground state reactions of acyl silanes under photochemical conditions.”

(2) Are the authors able to confirm the emission wavelengths of the white LEDs used? How much of the emission band overlaps with the absorption of the acyl silane - it appears unusual that only 1mol% of photocatalyst is sufficient to quench the acyl silane absorption which would indicate that there is not a huge amount of emission at from the white LED that corresponds to the absorption of the acyl silane.

Response: Thanks for the comment. As can be seen from the spectra, there is not a huge amount of the emission band overlaps with the absorption of the acyl silane, although white light indeed

induced the decomposition of acylsilanes (Fig 2 of our manuscript). The energy transfer from excited acylsilanes to Eosin Y probably inhibited the decomposition of the acylsilanes.

Reviewer #2 (Remarks to the Author):

This article by Shen and coworkers describes an efficient photocatalytic synthesis of cyclopentanol derivatives from the reaction of alkene-tethered acylsilanes with allylic sulfones in the presence of Eosin Y under blue irradiation. The broad scope of the substrate, along with the high reaction efficiency, are excellent features of this method. The study is well conducted, the SI is complete, and the results clearly presented. I think this work deserves to be considered for publication in Nature Comm. However, some important mechanistic concerns should first be addressed before acceptance of this work.

Response: Thanks for the positive recommendation.

1) As stated by the authors, one of the most challenging aspects of their reaction is the inhibition of the photochemical decomposition of acylsilanes by using photosensitizers with low triplet energy. In this context, Eosin Y (ET = 43.6 kcal/mol) was selected as an ideal photocatalyst. However, this problem can also be circumvented by simply irradiating at 530 nm as at this wavelength only Eosin Y will be excited. Looking at the supporting information (Table S2; entry 5), I was surprised to see that the reaction didn't proceed under green light irradiation.

Response: Thanks for the comment. When the green light was used, the reaction proceeded but with slightly less efficiency than the reaction under white light (Table S2; entry 5 and entry 6). We think the white light is more readily available than green light, and it could provide light with broader energy area, so we chose white light to further optimization the reaction conditions. We think the successful inhibition of white light induced decomposition of acylsilanes might be more useful for the design of new reactions in the future.

2) The luminescence quenching experiment depicted in Figure 2 tells only that the excited state of the Eosine is quenched with different concentrations of 1a. However, it does not confirm the involvement of triplet energy transfer as claimed by the authors. I believe only transient spectroscopy experiments could confirm the occurrence of such a process.

Response: Thanks for the kind suggestion. We have conducted the transient absorption measurements to investigate possible triplet-triplet energy transfer from 1a to Eosin Y. As

shown in the figure below, three different samples with (1) 1 mM **1a**, (2) 0.1 mM Eosin Y and (3) 1 mM **1a** + 0.1 mM Eosin Y were excited at 355 nm. Sample (1) with only 1 mM **1a** shows broad T1 - Tn transition at around 550 nm, although the signal was relatively weak due to the low molar absorption coefficient of triplet **1a**. Sample (3) with both **1a** and Eosin Y shows strong T1 - Tn transition of Eosin Y at around 580 nm, which is much stronger than the signal in sample (2) with only Eosin Y. The longer decay lifetime of **1a** + Eosin Y is also much longer than the triplet lifetime of **1a**. These results support is triplet-triplet energy transfer from **1a** to Eosin Y. We have added the transient absorption spectra into the revised Fig 2.

3) As a minor note, the authors should add a recent review on radical Brook rearrangements recently published in Angewandte Chemie (DOI: 10.1002/ange.202205671).

Response: Thanks for the kind suggestion. This review which was published online after we submitted the manuscript has been cited as reference 78 in the revised manuscript. The order of the other references has been updated accordingly.

REVIEWERS' COMMENTS

Reviewer #1 (Remarks to the Author):

Shen and co-workers describe the development of a new photochemical approach towards carbocyclic scaffolds that proposedly involves an unproductive energy transfer process between an acyl silane and photocatalyst.

The authors have adequately responded to the reviewers comments and concerns and have strengthened the manuscript through the inclusion of additional background information in the introduction in addition to transient absorption measurements that provide further support for the triplet-triplet energy transfer process. I now recommend publication.

Reviewer #2 (Remarks to the Author):

The authors have taken into consideration all criticisms and remarks raised in the first round of the evaluation. I believe the revised version is much better than the previous one and deserves to be published in Nat Commun

As a minor point, I would suggest to the authors add in the supporting information some experimental details about the laser flash photolysis used in this work.